# The Safety and Usefulness of Awake Surgery as a Treatment Modality for Glioblastoma: A Retrospective Cohort Study and Literature Review

**DOI:** 10.3390/cancers16152632

**Published:** 2024-07-24

**Authors:** Sho Osawa, Yasuji Miyakita, Masamichi Takahashi, Makoto Ohno, Shunsuke Yanagisawa, Daisuke Kawauchi, Takaki Omura, Shohei Fujita, Takahiro Tsuchiya, Junya Matsumi, Tetsufumi Sato, Yoshitaka Narita

**Affiliations:** 1Department of Neurosurgery and Neuro-Oncology, National Cancer Center Hospital, Tokyo 104-0045, Japan; soosawa@ncc.go.jp (S.O.); yasuji.miyakita@jfcr.or.jp (Y.M.); masataka@ncc.go.jp (M.T.); mohno@ncc.go.jp (M.O.); shuyanag@ncc.go.jp (S.Y.); dakawau2@ncc.go.jp (D.K.); tomura@ncc.go.jp (T.O.); s8o59.so3@gmail.com (S.F.); mephymach@gmail.com (T.T.); 2Department of Anesthesiology, National Cancer Center Hospital, Tokyo 104-0045, Japan; jmatsumi@ncc.go.jp (J.M.); tesatoh@ncc.go.jp (T.S.)

**Keywords:** awake surgery, glioblastoma, mapping result, extent of resection, complication

## Abstract

**Simple Summary:**

Awake surgery is the gold standard for localizing brain function and contributes to the maximal safe resection of brain tumors. On the other hand, the effectiveness of awake surgery for glioblastomas is controversial. One reason is that it is unclear whether awake surgery can be performed safely and whether functional areas can be detected in glioblastomas, which cause more severe edema than lower-grade gliomas. The purpose of this study was to examine the efficacy and safety of awake surgery for glioblastomas and to determine its current status through a literature review. Our study revealed awake mapping was successfully completed in 88%, and a positive response to mapping was observed in 53% of participants. The extent of resection and neurological deficits were comparable to previous studies. We concluded that awake surgery for glioblastomas can be safely performed and is useful for detecting functional areas. These findings influence treatment strategies for glioblastomas and improve treatment outcomes.

**Abstract:**

Awake surgery contributes to the maximal safe removal of gliomas by localizing brain function. However, the efficacy and safety thereof as a treatment modality for glioblastomas (GBMs) have not yet been established. In this study, we analyzed the outcomes of awake surgery as a treatment modality for GBMs, response to awake mapping, and the factors correlated with mapping failure. Patients with GBMs who had undergone awake surgery at our hospital between March 2010 and February 2023 were included in this study. Those with recurrence were excluded from this study. The clinical characteristics, response to awake mapping, extent of resection (EOR), postoperative complications, progression-free survival (PFS), overall survival (OS), and factors correlated with mapping failure were retrospectively analyzed. Of the 32 participants included in this study, the median age was 57 years old; 17 (53%) were male. Awake mapping was successfully completed in 28 participants (88%). A positive response to mapping and limited resection were observed in 17 (53%) and 13 participants (41%), respectively. The EOR included gross total, subtotal, and partial resections and biopsies in 19 (59%), 8 (25%), 3 (9%), and 2 cases (6%), respectively. Eight (25%) and three participants (9%) presented with neurological deterioration in the acute postoperative period and at 3 months postoperatively, respectively. The median PFS and OS were 15.7 and 36.9 months, respectively. The time from anesthetic induction to extubation was statistically significantly longer in the mapping failure cohort than that in the mapping success cohort. Functional areas could be detected during awake surgery in participants with GBMs. Thus, awake mapping influences intraoperative discernment, contributes to the preservation of brain function, and improves treatment outcomes.

## 1. Introduction

Awake surgery is an essential treatment modality in brain tumor surgery, preserving the neuronal function and maximizing the tumor removal of the eloquent areas of the brain. In particular, awake surgery is useful in the treatment of gliomas that invade normal tissues. Moreover, the indications thereof are expanding [1,2]. Recently, awake surgery as a treatment modality for glioblastomas (GBMs) has been shown to improve the extent of resection (EOR) and reduce complication rates [3,4], contributing to a longer overall survival (OS) [5,6]. However, additional studies exist regarding awake surgery possessing no advantage over general anesthesia. Furthermore, the usefulness thereof depends on the age and Karnofsky Performance Status (KPS) score of the patient [7,8,9]. No studies exist on the success rate of mapping or with positive findings of awake surgery as a treatment modality for GBMs. Therefore, the usefulness of awake surgery as a treatment modality for GBMs remains controversial. In this study, we determined the safety and usefulness of awake surgery as a treatment modality for GBMs, with a review of the literature.

## 2. Materials and Methods

### 2.1. Ethical Considerations

All participants gave their informed consent for inclusion before they participated in this study. This study was conducted in accordance with the Declaration of Helsinki, and the protocol was approved by the Research Ethics Review Committee of our hospital (Research Project number: 2013-042).

### 2.2. Study Design

Participants with either GBM, isocitrate dehydrogenase (IDH)-wildtype, World Health Organization (WHO) grade 4; astrocytoma, IDH-mutant, WHO grade 4; or GBM, not otherwise specified (NOS), WHO grade 4, who had undergone an awake craniotomy for tumor resection at our hospital, between March 2010 and March 2023, were included in this study. Those with recurrence were excluded from this study in order to make the clinical backgrounds as homogeneous as possible. All participants had undergone postoperative radiotherapy with concomitant and adjuvant temozolomide. This study included cases in which the tumor was located within or adjacent to the language areas of the cortex, language-related association fibers, primary motor cortex, primary somatosensory cortex, or premotor cortex. With regard to language function, awake surgery is indicated when the patient is able to perform some language tasks in a preoperative language function test. Regarding motor function, patients with no paralysis or mild paralysis who are able to perform social activities are considered eligible for awake surgery. During the study period, 91 awake surgeries were performed, of which 59 recurrent cases were excluded, and 32 primary diagnosed glioblastomas were included in this study.

### 2.3. Anesthesia

Anesthesia was administered, as per the Japanese Guidelines for Awake Surgery [10]. Anesthesia was induced using propofol in combination with remifentanil. Local anesthetics were administered around the pin fixation and skin incision sites. In addition, selective nerve blocks, involving the bilateral supraorbital, auriculotemporal, and greater and lesser occipital nerves, were performed. A supraglottic airway device was used for airway maintenance under anesthesia. Some participants were sedated with dexmedetomidine after task completion. Wound closure was performed under awake conditions.

### 2.4. Tasks

All participants underwent a preoperative evaluation of language function to select tasks that could be performed. Counting, visual naming, and auditory comprehension were performed as language tasks depending on the participant’s condition. Voluntary movements of the upper or lower extremities were observed in participants who had tumors adjacent to the primary motor cortex, premotor cortex, or pyramidal tract. Bipolar stimulation was performed at 2–10 mA for cortical and subcortical lesions. If language or motor arrest occurred during tumor removal, the cessation of the tumor removal occurred, and bipolar stimulation was performed along the removal site to determine the proximity thereof to the functional area. Mapping failure was defined as the inability to perform tasks intraoperatively.

### 2.5. Variables, Outcomes, and Definitions

The following variables were analyzed: age, sex, and the KPS score at admission and 3 months postoperatively; tumor site, maximum diameter; pathological diagnosis; IDH 1 or 2 status; response to intraoperative stimulation; the findings of response mapping; the EOR; convulsive seizure development, intraoperatively or ≤1 week postoperatively; postoperative neurological deficits; operative time; anesthetic time; time from anesthetic induction to extubation; time from extubation to the termination of the task; OS; and progression-free survival (PFS).

Tumor volumes were measured using preoperative and postoperative longitudinal relaxation time (T1)-weighted gadolinium contrast-enhanced magnetic resonance imaging (MRI). The EOR was calculated as follows: “(preoperative tumor volume—postoperative tumor volume)/preoperative tumor volume × 100” [11]. The EOR was classified as follows: gross total resection (GTR), (EOR 100%); subtotal resection (STR), (EOR 95%≤, <100%); partial resection (PR), (EOR 5%≤, <95%); and biopsy (collection of tissue for diagnosis only) [12]. In cases of the absence of a contrast effect, the calculation was based on the fluid-attenuated inversion recovery of hyperintense lesions.

Postoperative neurological deficits were categorized as early and late neurological deficits depending on whether they resolved within 3 months of surgery or lasted for >3 months postoperatively, respectively [13].

PFS was defined as the period from diagnosis to recurrence, mortality, or the last follow-up. OS was defined as the period from diagnosis to mortality or the last follow-up.

### 2.6. Statistical Analysis

The OS and PFS were analyzed using Kaplan–Meier survival curves. Survival times were presented as medians with 95% confidence intervals (CIs). Continuous variables were presented as medians and interquartile ranges (IQRs). Differences in the median values of the continuous variables were examined using the Mann–Whitney U test. Differences in the nominal variables were examined using Fisher’s exact test. Statistical analyses were conducted using EZR software (Version 1.61) [14]. Statistical significance was set at *p* < 0.05.

## 3. Results

### 3.1. The Demographic and Clinical Characteristics of the Participants

The data of 32 participants were analyzed. The clinical characteristics of these participants are presented in Table 1. The median age was 57 (45–68) years old, 17 (53%) of the participants were male, and the KPS score at admission was 80 (80–90).

The pathological diagnoses included 27 cases (84%), 4 cases (13%), and 1 case (3%) of GBM, IDH-wildtype; astrocytoma, IDH-mutant; and GBM, NOS, respectively. The dominant language area was involved in all cases. Moreover, the lesion was localized to the left and right hemispheres in 29 cases (91%) and 1 case (3%), respectively. Moreover, a butterfly glioma was present in two cases (6%). The locations included the frontal, temporal, and parietal lobes in 14 cases (44%), 12 cases (38%), and 2 cases (6%), respectively, and were multilobular in 4 cases (13%).

### 3.2. Anesthetic and Intraoperative Results

The median operative time was 416 (366–462) min, and the median time from the induction of anesthesia to extubation was 196 (167–217) min. The median time from extubation to the completion of the task was 199 (140–235) min, and the median time from the induction of anesthesia to the end of surgery was 518 (475–560) min.

### 3.3. Adjuvant Therapy

The adjuvant therapies administered postoperatively are presented in Table 2. All participants received radiotherapy with concomitant temozolomide. Postoperatively, adjuvant temozolomide, bevacizumab, and other drugs were administered to 32 (100%), 21 (66%), and 10 (31%) participants, respectively. Tumor-treating fields were applied in 7 participants (22%), while reirradiation and repeat surgeries were performed in 10 participants (31%).

### 3.4. Surgical Outcomes

The surgical outcomes are shown in Table 3. Awake mapping was successfully completed for 28 participants (88%); however, the tasks could not be performed or evaluated for 4 participants (13%).

In two cases, general anesthesia was reinduced due to insufficient wakefulness, and the tumors were removed under general anesthesia. Language and motor mapping were successfully completed in 28 and 2 participants, respectively. Positive responses to awake mapping were observed in 17 participants (53%), with language and motor responses in 17 and 2 participants, respectively. Of the participants with positive responses to awake mapping, 10 (31%) and 11 (34%) showed a positive response to cortical and white matter stimulation, respectively.

The resection of mapping-positive sites was not performed in 13 participants (41%). The EOR was categorized as GTRs, STRs, PRs, and biopsies, in 19 cases (59%), 8 cases (25%), 3 cases (9%), and 2 cases (6%), respectively.

Intraoperative seizures occurred in two participants (6%); however, cooling with artificial cerebrospinal fluid resulted in the cessation of seizures in both participants. Seizures occurred within 1 week postoperatively in three participants (9%); nonetheless, status epilepticus did not occur in any participants.

Early neurological deficits were observed in eight participants (25%). Conversely, only three participants (9%) presented with late neurological deficits. The late neurological deficits included hemiparesis in two cases and an oculomotor palsy of an unknown etiology in one case. Five participants presented with transient aphasia, sensory impairment, and acalculia during the acute postoperative period.

Among the 13 participants in whom the resection of the mapping-positive site was not performed, only 1 participant presented with late neurological deficits, due to a delayed cerebral infarction. The median KPS score at 3 months postoperatively was 90 (80–90). Neither postoperative pneumonia nor deep venous thrombosis was observed.

### 3.5. Mapping Failure

Factors correlated with mapping failure due to insufficient consciousness are shown in Table 4. The time from anesthetic induction to extubation was statistically significantly longer in the mapping failure cohort than that in the mapping success cohort (274 and 183 min, respectively; *p* = 0.002). The preoperative KPS score ≦ 80 (100% and 50%, respectively; *p* = 0.113), age (54 and 57 years, respectively; *p* = 0.711), male sex (50% and 54%, respectively; *p* = 1), and maximal diameter (50 and 49 mm, respectively; *p* = 0.732) did not statistically significantly differ between the cohorts.

### 3.6. PFS and OS

The median PFS and OS were 15.7 months (95% CI: 9.9–21.2) and 36.9 months (95% CI: 28.4–49.6), respectively (Figure 1A,B). The median PFS and OS of participants with IDH-wildtype GBMs were 11.1 months (95% CI: 9.6–17.8) and 31.8 months (95% CI: 26.3–not available [NA]), respectively (Figure 1C,D).

The survival curves, based on the EOR, are depicted in Figure 2. The PFS was 12.9 (95% CI: 8.5–21.2), 24.0 (95% CI: 7.6–NA), and 13.9 (95% CI: 9.7–NA) months for a GTR, STR, and PR or biopsy, respectively (*p* = 0.421; Figure 2A). The OS was 36.9 (95% CI: 17.3–NA), 40.8 (95% CI: 28.4–NA), and 30.7 (95% CI: 13.1–NA) months for a GTR, STR, and PR or biopsy, respectively (*p* = 0.594, Figure 2B).

The survival curves of IDH-wildtype GBMs, based on the EOR, are depicted in Figure 2C,D. The PFS was 11.1 (95% CI: 8.3–21.6), 17.8 (95% CI: 7.6–NA), and 9.9 (95% CI: 9.7–NA) months for a GTR, STR, and PR or biopsy, respectively (*p* = 0.809; Figure 2C). The OS was 36.9 (95% CI: 17.3–NA), 40.7 (95% CI: 28.4–NA), and 30.7 (95% CI: 13.1–NA) months for a GTR, STR, and PR or biopsy, respectively (*p* = 0.595, Figure 2D).

Of the 32 participants who underwent awake surgery, 11 underwent a secondary surgery. All secondary surgeries were performed while the participant was awake. The EOR of these secondary surgeries included GTRs, STRs, and PRs in four cases, three cases, and four cases, respectively. A pathological diagnosis of 1 case of PR requiring a second surgery was radiation necrosis, while the other 10 cases were due to recurrence.

## 4. Discussion

Awake surgery for GBMs was successfully performed in 28 participants (88%). A positive response to mapping was observed in 17 participants (53%), and the resection of the mapping-positive sites was not performed in 13 participants (41%). These findings revealed that awake surgery as a treatment modality for GBMs was safely performed and influenced intraoperative discernment. Furthermore, the time from anesthetic induction to extubation was correlated with successful awake mapping.

The EOR of GBM is associated with the prognosis thereof [15,16,17,18,19,20]. Awake surgery for gliomas is reportedly effective for maximal safe resection [1]. However, the usefulness and safety of awake surgery as a treatment modality for GBMs remain controversial. Moreover, due to potential severe edema and aggressive invasiveness, whether awake surgery as a treatment modality for GBMs can be safely performed and whether positive findings can be obtained have not been previously elucidated. Our study revealed that awake surgery as a treatment modality for GBMs can be performed safely while detecting functional areas. To the best of our knowledge, this is the first study to report the success and positive response rates of awake mapping for GBMs.

We reviewed the literature on awake surgery as a treatment modality for GBM, from 2009 to 2023, identifying eight studies. A summary of these previous studies on awake surgery as a treatment modality for GBM is presented in Table 5. Gerritsen et al. and Li et al. have observed that awake surgery increases the EOR and prolongs the PFS or OS of patients with GBMs [3,5,6]. Conversely, Gallet et al. and Fukui et al. have found that awake surgery does not increase the EOR or prolong the PFS or OS of patients with GBMs [7,8]. The findings of a study by Nakajima et al. have revealed that awake surgery as a treatment modality for GBM is useful for preserving the KPS score; however, this is dependent on age and the preoperative KPS score [9]. Moreover, Gerritsen et al. have demonstrated that the usefulness of awake surgery as a treatment modality for GBM is uncertain in patients who are >70 years old, have a preoperative National Institutes of Health Stroke Scale score of ≥2, or have a preoperative KPS score of ≤80 [5]. Further studies are needed to establish the usefulness or detect sub-cohorts that benefit from awake surgery as a treatment modality for GBM. Prospective cohort studies and randomized controlled trials are currently underway to evaluate the efficacy of awake surgery as a treatment modality for GBMs and high-grade gliomas [21,22].

Gerritsen et al. have found that awake surgery as a treatment modality for GBMs reduces neurological deficits at 3 months postoperatively, while Gallet et al. have observed no significant differences in neurological deficits resulting from awake surgery when compared to that of general anesthesia at 1 month postoperatively (Table 5) [3,5,8]. The incidence of neurological deficits is 20–27% and 8.1–32.6%, at 1 month and 3 months postoperatively, respectively (Table 5) [3,5,8,23,24]. In our study, the incidence of neurological deficits lasting >3 months postoperatively was 9%, which is comparable to that reported in previous studies. The detection of functional lesions is associated with good functional outcomes. Because postoperative neurological deficits are associated with a poor prognosis of GBMs [25], maximal safe resection under awake mapping is an ideal treatment strategy for GBMs located within or adjacent to the eloquent areas. To improve the prognosis of glioblastoma, we believe that it should be treated at high-volume centers where multidisciplinary care, including awake surgery, can be performed.

**Table 5 cancers-16-02632-t005:** Summary of included studies.

Author, Year	Sample Size	EOR	Neurological Deficits	OS/PFS
Gerritsen, 2019 [3]	AC 37 vs. GA 111	mean EOR: AC 94.89% vs. GA 70.30% (*p* = 0.0001)	Minor deficits 3 months after surgery: AC 3% vs. GA 15% (*p* = 0.05) Major deficits 3 months after surgery: AC 5% vs. GA 12% (*p* = 0.27)	median OS: AC 17 months vs. GA 15 months (*p* = 0.27)
Gerritsen, 2022 [5]	AC 134 vs. GA 402	mean EOR AC 95·4% vs. GA 86·3% (*p* < 0·0001)	3 months after surgery: AC 22% vs. GA 33% (*p* = 0.019)	median PFS: AC 9.0 months vs. GA 7.3 months (*p* = 0.0060) median OS: AC 17.0 months vs. GA 14.0 months (*p* = 0.00054)
Fukui, 2022 [7]	AC 15 vs. GA 15	mean EOR AC 99.5% vs. GA 97.9% (*p* = 0.231)	NR	median OS: AC 30.4 months vs. GA 16.0 months (*p* = 0.381)
Li, 2021 [6]	AC 48 vs. GA 61	EOR ≧ 95%: AC 83.3% vs. GA 45.9% (*p* < 0.0001) mean EOR: AC 94.9% vs. GA 90.2% (*p* = 0.003)	NR	mean PFS: AC 23.2 months vs. GA 18.9 months (*p* = 0.001) mean OS: AC 28.1 months vs. GA 23.4 months (*p* < 0.001)
Nakajima, 2021 [9]	AC 30 vs. GA 30	mean EOR: AC 97.0% vs. GA 96.0% (ns)	NR	NR
Gallet, 2022 [8]	AC res 36 vs. GA res 37 vs. GA bx 14	EOR ≧ 90%: AC 50% vs. GA 51% (*p* = 1.000)	1 month after surgery Language: AC res 19% vs. GA res 11% vs. GA bx 21% (*p* = 0.099) Motor: AC res 8% vs. GA res 5% vs. GA bx 36% (*p* < 0.001)	median PFS: AC res 7.3 months vs. GA res 11.6 months vs. GA bx 7.8 months (*p* = 0.285) median OS: AC res 17.5 months vs. GA res 23.4 months vs. GA bx 21.8 months (*p* = 0.650)
Kim, 2009 [24]	AC 137	EOR ≧ 95%: 73%	1 month after surgery: 20%	NR
Clavreul, 2021 [23]	AC 46	EOR 100%: 61% EOR 90–99%: 33%	3 months after surgery: 32.6%	median PFS 6.8 months median OS 17.6 months
Present study	AC 32	EOR 100%: 59% EOR 95–99%: 25%	Within 3 months after surgery: 25% Lasted over 3 months after surgery: 9%	median PFS 15.7 months median OS 36.9 months

AC = awake craniotomy, bx = biopsy, EOR = extent of resection, GA = general anesthesia, NR = not reported, OS = overall survival, PFS = progression-free survival, res = resection.

The STR group paradoxically tended to have a better prognosis than the GTR group, although the difference is not statistically significant. In the STR group, a positive response to mapping was observed in seven of the eight participants, and removal was discontinued in six cases. As a result, there was no late deficit, and the KPS score 3 months postoperatively was 90 in seven of the eight participants. On the other hand, 3 of the 16 participants in the GTR group developed a late deficit, and the postoperative KPS score tended to be poor, with 11 of the 19 participants having a KPS score of 90 and the other 8 participants having 80 or less (*p* = 0.201). The differences in the late neurological deficits and postoperative KPS scores as a consequence of preserved neuronal function in the STR group might reflect the outcome.

The median OS of patients with GBMs in Japan is 18 months [26]. In the “randomized screening phase II trial of interferon β plus temozolomide in comparison with temozolomide alone for newly diagnosed GBM” (JCOG0911 study), in patients with a KPS score ≥ 70, the median OS after diagnosis with a GBM was found to be approximately 20 months [27]. Moreover, a previous study found that the median or mean OS of patients with GBMs who underwent awake surgery was 17.0–30.4 months (Table 5). In the present study, the median OS was 36.9 months, which was the most superior result when compared to those of previous studies [3,5,6,7,8,23]. Although the WHO classification of tumors of the central nervous system changed during the study period, the prognosis of glioblastoma, IDH-wildtype, which met the current diagnostic criteria, was comparable to that of the entire cohort. Maximal safe resection under awake mapping contributes to a good prognosis; nevertheless, a high KPS score, young age, and differences in postoperative treatment could additionally contribute to the prognosis. In particular, bevacizumab and repeat surgery, which contribute to prolonging the survival of GBM, may have influenced the outcomes.

Insufficient wakefulness in awake surgery occurs in 5.2–19% of cases. Moreover, insufficient wakefulness in awake surgery is associated with an age of ≥70 years old, uncontrolled epileptic seizures, previous oncological treatment, hyperperfusion on MRI, a mass effect on the midline, and a left-sided lesion [10,28,29].

Age, a non-smoking status, an American Society of Anesthesiologists class III, IDH-wildtype tumors, and repeated surgeries are associated with delayed awakening from an awake surgery [30,31]. Additionally, an association between a reduced preoperative function of the parietal lobe and intraoperative consciousness has been shown [32]. In our study, 13% of participants could not undergo awake mapping due to insufficient wakefulness. This incidence is comparable to that reported in previous studies, revealing the feasibility of this technique for GBMs. Moreover, we demonstrated that the time from anesthetic induction to extubation was statistically significantly longer in the patients in the mapping failure cohort. Thus far, the correlation between anesthetic time and mapping failure has not yet been reported. Therefore, further studies are required to confirm the validity of these results.

This study had a limitation, due to being a retrospective analysis. Therefore, our results require validation by larger studies.

## 5. Conclusions

Functional areas could be detected during awake surgery in participants with GBMs. These results revealed that awake mapping influences intraoperative discernment, contributes to the preservation of brain function, and improves treatment outcomes. As the time from anesthetic induction to extubation affects the success of mapping, the prompt preparation of surgery and surgical procedures is necessary.

## Figures and Tables

**Figure 1 cancers-16-02632-f001:**
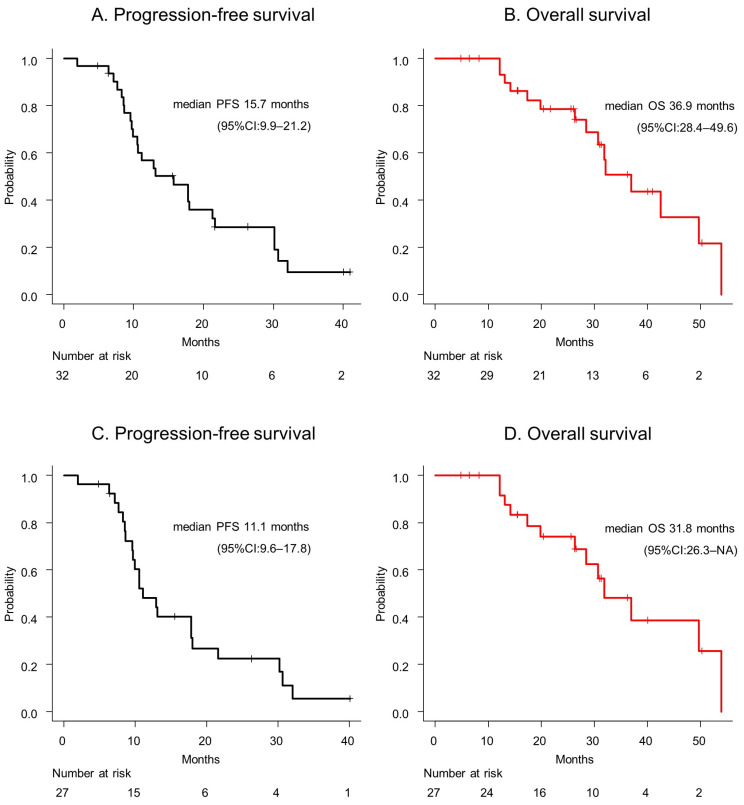
(**A**) The progression-free survival and (**B**) overall survival of a glioblastoma patient are presented. (**C**) The progression-free survival and (**D**) overall survival of a glioblastoma, IDH-wildtype patient are depicted. Abbreviations: CI = confidence interval; OS = overall survival; PFS = progression-free survival.

**Figure 2 cancers-16-02632-f002:**
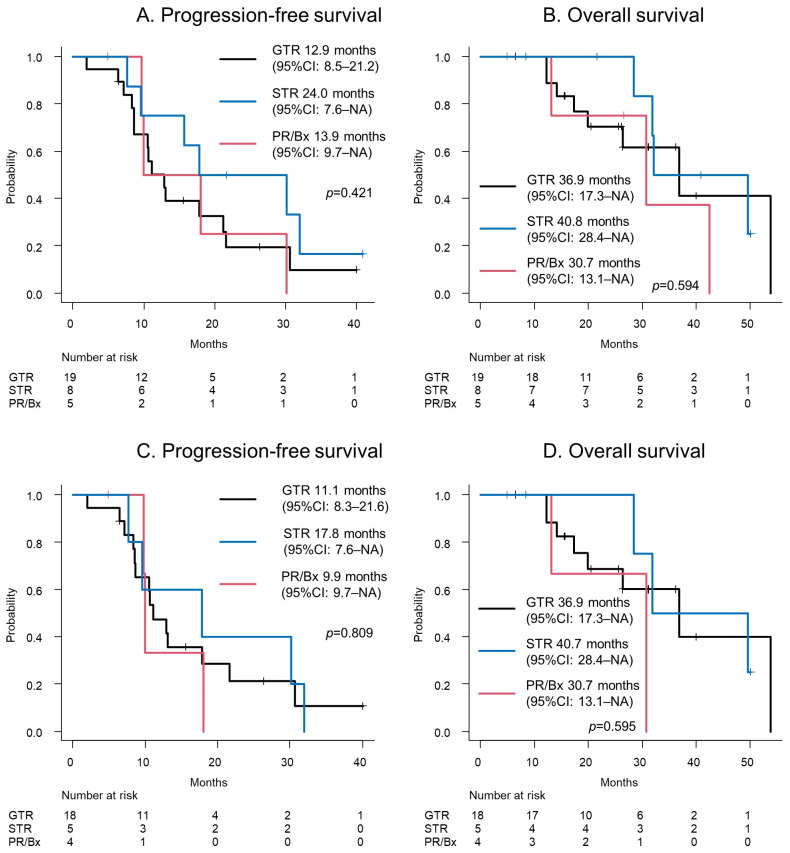
(**A**) The progression-free survival and (**B**) overall survival of a glioblastoma patient are depicted. (**C**) The progression-free survival and (**D**) overall survival of a glioblastoma, IDH-wildtype patient are shown. Abbreviations: Bx = biopsy; GTR = gross total resection; NA = not available; PR = partial resection; STR = subtotal resection.

**Table 1 cancers-16-02632-t001:** Clinical characteristics of glioblastoma patients.

Characteristics		(% or IQR)
Age, years, median (IQR)	57	(45–68)
Men, (%)	17	(53)
KPS on admission	80	(80–90)
Pathological diagnosis		
Glioblastoma, IDH-wild	27	(84)
Glioblastoma, IDH-mutant/Astrocytoma grade 4	4	(13)
Glioblastoma, NOS	1	(3)
Location		
Frontal	14	(44)
Temporal	12	(38)
Parietal	2	(6)
Multilobular	4	(13)
Laterality		
Left	29	(91)
Right	1	(3)
Bilateral (butterfly glioma)	2	(6)

IDH = isocitrate dehydrogenase. IQR = interquartile range. KPS = Karnofsky Performance Status. NOS = not otherwise specified.

**Table 2 cancers-16-02632-t002:** Details of the treatment after surgery.

Treatments		(%)
Radiotherapy concomitant with TMZ	32	(100)
Adjuvant TMZ	32	(100)
Bevacizumab	21	(66)
Other drugs	10	(31)
Tumor-treating fields	7	(22)
Reirradiation	10	(31)
Repeat surgery	10	(31)

TMZ = temozolomide.

**Table 3 cancers-16-02632-t003:** Outcomes of awake surgery for glioblastoma patients.

Surgical Outcomes		
Awake mapping, (%)		
Successfully completed	28	(88)
Failure	4	(13)
Positive response to awake mapping, (%)		
Cortical mapping	10	(31)
Subcortical mapping	11	(34)
Response to mapping results, (%)		
Stop resection	13	(41)
EOR, (%)		
GTR	19	(59)
STR	8	(25)
PR	3	(9)
Bx	2	(6)
Seizure, (%)		
Intraoperative	2	(6)
Postoperative, acute periods	3	(9)
Early neurological deficits, (%)	8	(25)
Late neurological deficits, (%)	3	(9)
KPS 3 months after surgery, (IQR)	90	(80–90)
OS, months, (95%CI)	36.9	(28.4–49.6)
PFS, months, (95%CI)	15.7	(9.9–21.2)

Bx = biopsy. CI = confidence interval. GTR = gross total resection. IQR = interquartile range. KPS = Karnofsky Performance Status. OS = overall survival. PFS = progression-free survival. PR = partial resection. STR = subtotal resection.

**Table 4 cancers-16-02632-t004:** Factors associated with mapping failure.

	Failure (*n* = 4)		Successfully Completed (*n* = 28)		*p* Value
Age, years	54	(44–61)	57	(45–70)	0.711
Male	2	(50)	15	(54)	1
KPS ≦ 80	4	(100)	14	(50)	0.113
Maximal diameter, mm	50	(43–60)	49	(40–60)	0.732
Time from anesthetic induction to extubation, minutes	274	(246–296)	183	(163–208)	0.002

KPS = Karnofsky Performance Status.

## Data Availability

The data are not publicly available due to privacy restrictions.

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
