# Peer review of "The Safety and Usefulness of Awake Surgery as a Treatment Modality for Glioblastoma: A Retrospective Cohort Study and Literature Review"

_cancers, 2024, doi:10.3390/cancers16152632_

Round 1

Reviewer 1 Report

Comments and Suggestions for Authors

The present study assessed the apply of awake surgery as a treatment modality for glioblastomas (GBMs) and aims to analyze its efficacy, safety, and factors correlated with mapping failure. 

Awake surgeries have traditionally been performed for low-grade gliomas in functional areas. However, the question of whether this technique can be safely applied to GBM and whether it reduces progression-free survival (PFS) remains controversial. The methodology and results sections of the study provide valuable insights into the clinical outcomes of awake surgery in GBM patients. Nonetheless, there are a few points that could be improved upon in the current research.:

1.     A control group of GBM without awake surgery in the same center could be set in the study in order to improve the evidence level of the study. 

2.     Provide more details on the patient selection criteria and exclusion criteria. Explain why recurrent cases were excluded and how this affects the study population.

3.     Adjuvant therapy were administered postoperatively in deferent cases (table 2). Whether the different therapy strategy influence the result could be discussed in detail. 

4.     Clarify the definition of mapping success and failure. Provide more details on how mapping success or failure was determined.

Overall, the paper provides an interesting analysis of awake surgery for GBMs. With some revisions to clarify the objectives, strengthen the methodology, and expand on the discussion, the paper could make a valuable contribution to the field.

Comments on the Quality of English Language

Minor editing of English language required

Reviewer 2 Report

Comments and Suggestions for Authors

I agree with the authors in their statement that awake surgery is safe and useful for the treatment of astrocytoma grade 4 tumors. This has been extensively published and experienced by many oncological neurosurgeons in their daily practice. What is most important is that this type of surgery  results in safe minimization of postoperative tumor volume as well as improvement of postoperative KPS, depending on tumor location, to lengthen OS.

 This said, the work presented is not adhering to new terminology and classification systems of tumors and EOR, which adds confounders to their results. The long PFS and OS of the patients taking into account that only 59% had a GTR (whatever this means) and only 20% received RT (TMZ/RT protocol?). But what is more striking is that these impressive results are even better for STR than for GTR, which is difficult to understand.

 I will now extract my commentaries to the text downloaded:

 Line 16, 52: awake surgery certainly contributes to obtain maximal safe resection in tumors related to eloquent brain areas (as has been extensively pulished), Awake surgery is actually the gold standard for localizing functional areas in the brain, whenever you plan to preserve such areas.

Line 19: all gliomas are invasive tumors by nature, this is not a specific characteristic of grade 4 astrocytomas.

Line 43: PFS and OS results are much longer than those obtained in most series. ¿Is there any selection bias you can mention to understand this difference?

Line 76: during the years 2010-2023 there has been major changes in the WHO classification of brain gliomas. There is no mention to posible pathological confounders related to this fact.

Line 97: cortical and subcortical stimulation was performed

Line 110: EOR should be adapted to the last RANO classificationcited in the text and including both CE and non CE áreas of the tumor. The supramarginal resection is not even contemplated in the description in this paper,  although difficult to achieve in tumors located near eloquent areas.

It would also be important not only to define de EOR with respect to the percentage of tumor removed but also the measurement of the remaining tumor volumen, including the non CE tumor. Both measurements are relevant in terms of EOR and OS. This dual measurement has been also included in the new classification systems of EOR.

Line 117: should say “resolved” instead of “developed”.

Line 147: surgical timing seems quite long compared to other standards: 8 hours median per surgery, more than 3 hours from induction to the start of awake tasks, more than 3 hours of task work. If longer time of anestesia prior to wakening the patient for the verbal/motortask really influences negatively the collaboration of the patient, this should be kept in mind and be shortened.

Table 2: ¿Only 20% of the operated patients received postop radiotherapy in first instance?

Line 171: “Resection of the mapping-positive sites was not performed in 13 participants (40.6 %)”. Does this mean that, in others, the positive mapping áreas were resected?

The EOR was categorized as GTRs 19 cases (59.4%), and this is related only to the CE tumor, which means that we are not sure if this GTR really corresponds to the actual Class 2 maximal CE resection, class 2ª or 2B

Figure2: PFS and OS is better after obtaining a STR than a GTR? This is against the general experience of many oif us; the greater the safe resection, the better survival of the patient. This needs some explanation and maybe include opening the possibility to consult the data, which are unavailable.

I thank the authors for their effort suporting awake surgery even for malignant brain tumors, which for sure will help quality and quantity of survival for the patients. 

Reviewer 3 Report

Comments and Suggestions for Authors

Authors present a cohort study of patients with GBM who underwent awake surgery. This is an important point, especially in terms of achieving a good functional outcome. Patients with GBM form an extremely relevant cohort, given their limited life expectancy. I have some minor questions. If I understand correctly, the procedure involve asleep, awake and asleep phase. Was a supraglottic device used throughout the entire procedure, or was an endotracheal tube used later after the mapping? Did you see some complications here? I would appreciate it if you could provide more precise details on the selection criteria for patients undergoing awake surgery. Do you perform always awake surgery on lesions located adjacent to the motor cortex or adjacent to the somatosensory cortex or did you follow other criteria?

Reviewer 4 Report

Comments and Suggestions for Authors

Modern neurosurgical proposals include awake surgery as a relevant approach, in this field the text by Osawa, S. et. al., is important; however, the authors should clarify some points:

1.      The use of decimals as an approximate round value when the n=figure is 32 is unnecessary (e.g. 40.6% is 41%; 53.1% is 53%).

2.      Although the authors recommend awake surgery based on their results, this process is expensive; long and the effects (P9,L238-243) of GBM therapy are still dismay.

3.      I suggest that this point should be briefly discussed as many centers would have difficulties for awake surgery.

4.      Figure legends should be clarified (are C and D in figures 1 and 2 necessary?)
